# Structural Determinants of Child Health in Rural China: The Challenge of Creating Health Equity

**DOI:** 10.3390/ijerph192113845

**Published:** 2022-10-25

**Authors:** Yunwei Chen, Sean Sylvia, Sarah-Eve Dill, Scott Rozelle

**Affiliations:** 1Department of Health Policy and Management, Gillings School of Global Public Health, University of North Carolina at Chapel Hill, Chapel Hill, NC 27599, USA; 2Stanford Center on China’s Economy and Institutions, Stanford University, Stanford, CA 94305, USA

**Keywords:** health inequity, child health, rural China, review

## Abstract

Over the past two decades, the literature has shown a clear gradient between child health and wealth. The same health–wealth gradient is also observed among children in China, with a large gap in health between rural and urban children. However, there are still unanswered questions about the main causes of China’s rural–urban child health inequality. This paper aims to review the major factors that have led to the relatively poor levels of health among China’s rural children. In addition to the direct income effect on children’s health, children in rural areas face disadvantages compared with their urban counterparts from the beginning of life: Prenatal care and infant health outcomes are worse in rural areas; rural caregivers have poor health outcomes and lack knowledge and support to provide adequate nurturing care to young children; there are large disparities in access to quality health care between rural and urban areas; and rural families are more likely to lack access to clean water and sanitation. In order to inform policies that improve health outcomes for the poor, there is a critical need for research that identifies the causal drivers of health outcomes among children. Strengthening the pediatric training and workforce in rural areas is essential to delivering quality health care for rural children. Other potential interventions include addressing the health needs of mothers and grandparent caregivers, improving parenting knowledge and nurturing care, improving access to clean water and sanitation for remote families, and most importantly, targeting poverty itself.

## 1. Introduction

Over the past two decades, the literature has shown a clear “gradient between health and wealth”, where health tends to improve as wealth increases [1]. Children who grow up in poor families tend to have poorer health compared with those in wealthier families [2,3]. This pattern exists for a wide range of health outcomes, including general health ratings [3], functional limitations [4], growth [5,6], morbidity [4], and mortality [7]. In correlational studies of family income and child health (e.g., those studies in Fletcher and Wolfe’s review [3]), as well as studies that use some types of natural experiments [8], nearly all research teams find significant relationships with meaningful magnitudes.

The health–wealth gradient is also consistent across age ranges and in a wide variety of countries. These relationships have been found among children ages 0–3, 4–7, 8–12, and 13+ in several countries including the US [2,3,9], Canada [10], the UK [11,12], Australia [13], and Germany [14]. While most papers on the health–wealth gradient are written on high-income countries, papers in low- and middle-income countries (LMICs), for example, South Africa [15] and Mexico [8], also find significant positive effects of increasing family income on the health status of children. This relationship extends to cognitive development as well: In a study of children in five Latin American countries, Schady et al. find steep gradients between language acquisition and socioeconomic measures [16]. 

The same health–wealth gradient is also observed among children in China. Consistent with the evidence from other countries, Goode et al. found a strong health gradient by family income among children in China [6]. This gradient increased with child age, with greater effects of family income on children with poorer health and children living in rural areas. Poorer children in China were more likely to experience multiple illnesses, and poorer households were less likely to address health conditions effectively. Zhang et al. also found significant inequality in early child development between children in the top and bottom quintiles of the family income distribution [17].

These gradient relationships can perpetuate the intergenerational transmission of poverty and health in a vicious cycle. The effect of low family income can accumulate during early childhood and have lasting negative consequences on adult health and well-being [18]. Children born in poor families may suffer from both lower socioeconomic status and poorer health when transitioning into adulthood, triggering an intergenerational transmission of poor health and poverty [2]. Carvalho found that childhood health and development (including nutrition, cognitive skills, and noncognitive abilities) explain more than one-third of the intergenerational transmission of socioeconomic status between parents and their offspring [19].

Above all, given the high level of inequality that exists in China [20] and the fact that the rural–urban divide accounts for a large share of the income gap [21], it is not surprising that a number of studies show a persistent gap in health between rural and urban children. For example, Liu et al. found that rural children were more likely to be stunted or underweight than urban children during the 1989–2009 period, using the China Health and Nutrition Survey that covers nine provinces and 45% of China’s population [22,23]. Xu and Huang used six successive cross-sectional surveys from the Chinese National Survey on Student Constitution and Health to find a similar trend: great height inequalities existed between urban and rural children aged 7–18 years during the 1985–2010 period of their study [24]. Gao et al. found that socioeconomic status disparities in child growth persisted in China at least from 1991 to 2015 [25]. Finally, Rao et al. found persistent rural–urban gaps in early developmental outcomes during the 2010–2018 period [26]. Such urban–rural differences in child health and development are consistent with findings from other LMICs [5].

Despite the large gaps in health outcomes between urban children from relatively well-off families and rural children from relatively poorer families, the incomes of rural families have risen steadily over the past several decades. This trend, in part, appears to have contributed to rising levels of health outcomes in children. Indeed, from the early 1990s through the late 2010s, rural income per capita has risen by nearly 20 times [27]. Over this same period, data also show that the health of rural children has improved considerably. Infant mortality rates in rural areas decreased from 58 cases per 1000 live births in 1990 to 6.6 cases per 1000 in 2019 [28], compared to a global decline of 71 to 31 cases per 1000 in LMICs over the same period. The under-5 mortality rate in rural areas has decreased from 71.1 per 1000 in 1990 to 9.4 per 1000 in 2019 [28], compared to a global decline of 103 to 41 in LMICs globally. Rural children’s height-for-age and weight-for-age z-scores have increased by 66% and 129%, respectively, and stunting and underweight prevalence rates have correspondingly decreased by approximately 37% and 60% from 1989 to 2006 [23].

In addition to the rapid socioeconomic development, several recent policy efforts in China may have contributed to improvements in the health status of rural children. The development of a hierarchical primary care and public health system from the national level to the village level aimed to increase the provision of clinical treatment and maternal and child health services in the rural areas. Several national maternal and child health programs since the 2009 national reform have also ensured the coverage of a wide range of maternal and child health services, including immunizations, child health management, antenatal and postnatal health care, and nutrition improvement [29]. These national policies, along with financial support from the central government, the coverage of social health insurance, improvements in health infrastructure and the healthcare workforce, and the implementation of the national surveillance system may have also contributed to improving health outcomes among rural children [28].

Despite the positive changes that have occurred in the rural health system, there remains significant inequality in health outcomes between urban and rural children [3]. There are also unanswered questions about the main causes of China’s rural–urban child health inequality. Identifying and targeting the sources of inequality is important given the lasting effects of childhood health and development on adult health [18,30], education attainment [31], cognition achievement [32], adult economic activities [33], and adult socioeconomic status [19].

The purpose of this paper is to review the major factors that have been discussed in the literature to explain the relatively poor levels of health among China’s rural children. Although the relationship between socioeconomic status and health is well documented in the literature, the reasons for this relationship are far from clear (because causal mechanisms may plausibly move in both directions and because potential third factors may simultaneously affect socioeconomic status and health). Starting from the work by Case et al. [2], many studies focusing on children assume away the plausible channel that runs from child health to socioeconomic status as child health may have relatively little impact on their own socioeconomic status. However, this strategy does not rule out many possible third factors explanations, such as the intergenerational transfer of health and socioeconomic status, the lasting effect of poor health at birth, the accumulation of weak health in early ages, and poor access to quality care. In next section, we lay out a few possible mechanisms underlying the health gaps of rural children in China, following the framework by Case et al. on the wealth-health gradient among children [2].

## 2. What Underlies the Health Gaps of Rural Children?

### 2.1. The Causal Effect of Family Income on Children’s Health

First, family income may have a direct causal effect on children’s health as poor families are less able to purchase nutritious foods and other health inputs for their children. In light of the empirical difficulties associated with estimating causal effects, it is uncommon for studies to address the question of whether the estimated relationship is causal. Recent efforts have used natural experiments to quantify the direct effects of family income on child health outcomes. International examples include the experimental programs offering conditional cash transfers in Mexico [8,34] and Ecuador [35]. By exogenously increasing the family income through an experiment offering cash transfer, these studies have generally found that increasing the family income leads to improvements in the health status of children in poor families [36]. Another similar income-support program is to provide old-age pensions for the elderly. Such programs directly transfer cash to the elderly and their families, which may also have intergenerational effects on children. For example, Duflo and Case examined South Africa’s pension program and found a positive effect of program participation on child growth [15,37].

The above evidence from the international literature demonstrates the direct income effect on children’s health and suggests one possible channel of explaining the poor health of children living in rural areas of China. Consistent with the international literature, studies from China mostly came to a similar conclusion. Among them, Zheng et al. relied on the implementation of a pension scheme for rural residents that exogenously increased the family income of the rural elderly [38]. As we will discuss below, many children in rural areas live under the care of grandparents. The pension program is expected to increase the family’s income when grandparent caregivers receive the pension payments. The study found a significant effect on the health status of children and identified an increase in the consumption of high-protein food by children as the main mechanism by which higher incomes affected child health [38].

Although it appears from the literature that family income directly affects the health of children, this is only one possible explanation for the poorer health status of rural children in China. We note that explaining the observed income-health gradient associations is a complex task since there clearly is a presence of many other behavioral, social, and biological factors that may jointly affect both family income and child health. In fact, these factors may either reinforce or mitigate the negative effects of a family’s low income over time, generating cumulative and dynamic effects on child health in the long-term [39]. In the following, then, we discuss several major third-factor explanations that have been discussed in the literature.

### 2.2. Disparity in Child Health at Birth

Another possible mechanism underlying the health gaps of rural children in China could be disparities in child health at birth. Children in rural areas or from poor families could be at greater risk of being born with health problems due to inadequate prenatal care or other maternal characteristics associated with poverty. Such health problems may include low birth weight, prematurity, or birth defects. Poor birth outcomes could cause a disparity in health among young children and have lasting effects on health at later ages.

China has made outstanding progress in reducing maternal and child mortality over the past few decades. However, profound disparities in maternal and childcare delivery persist between rural and urban areas. For example, despite narrowing urban–rural differences between 1990 and 2019, the infant mortality rate in rural areas was still 1.9 times higher than in urban areas in 2019 [28]. A part of this disparity could stem from differences in the maternal and child health systems implemented in urban and rural areas [40]. China has implemented a comprehensive maternal and child health care program to promote prenatal health care in urban areas since 1987, while a similar program was only initiated in rural areas in 2000. Although the national initiative covered all poor rural areas by 2005 (which has been shown to have substantially improved prenatal care in rural areas), research shows that significant variations exist across regions [41,42,43,44], and prenatal care in some poor rural areas is still below the national average [43]. Other studies, mainly conducted in western China where most of the minority ethnic populations reside, found that although caregivers in some rural regions do receive some amount of perinatal care during pregnancy and after birth, large shares do not attend the recommended number of prenatal or postnatal care visits [42,45]. Prenatal care was most often inadequate for ethnic minorities and less educated women [41]. Another example is the inadequate nutrition of pregnant women, which is also correlated with poverty and child health at birth. A study by Ma et al. found a high prevalence of inadequate dietary diversity during pregnancy in rural China, which has been posited as a cause of high rates of anemia in young infants [46,47].

The above evidence has demonstrated that poor prenatal care among rural pregnant women may be a contributor to the lower health outcomes of rural neonates, generating cumulative adverse effects on health at later ages. At the same time, we acknowledge that these factors during pregnancy do not fully account for birth outcome disparities. For example, recent research has proposed a life-course approach to addressing early life disadvantages, arguing that disparities in birth outcomes are not just a result of factors that occur during the nine months of pregnancy but also from the entire life course of the mother prior to pregnancy [48,49]. Rural women may experience disproportionate health costs as a result of social and economic inequalities that they have experienced over the course of their lives, and these experiences may be contributing to the urban–rural disparities that are observed among the health of newborns.

### 2.3. Poor Health of Rural Caregivers

Poorer child health in rural areas may also be affected by the health status of their parents or caregivers. The first possible pathway is through the intergenerational transmission of poor parental health. For example, Eriksson et al. estimated the intergenerational transmission of health in China and found that 15% to 27% of the rural–urban disparity in child health can be derived from the inequality in the health of their parents [50]. When rural parents are in poor health, this decreases the socioeconomic status of rural children and is then associated with an increased risk of illness and can cause delays in early childhood development. Another possible pathway is through the poor care provided by less healthy or weaker caregivers. One example is the prevalent perinatal mental health problems among rural mothers in China. These problems have received little attention given the inadequate surveillance system for auditing mental health in China [51]. Jiang et al. found that depression, anxiety, and stress symptoms among rural mothers were significantly associated with reduced hand washing practice and a higher probability of infant morbidity [52].

In addition, the high volume of rural–urban migration in some regions of China (coupled with China’s strict household registration system) has resulted in a large number of rural children under the care of grandparent caregivers who often have poor health or are living with chronic conditions [53,54]. In China, rapid urbanization has contributed to the increased number of young parents who migrate to cities for work and leave their children in rural villages. Although migrant parents wish to take their children with them, the household registration system in China limits the access of rural migrants to urban resources including education, health care, and other social programs, creating barriers to family migration even if parents would like to keep their families together [53]. As of 2010, “left behind children (LBC)” numbered 61 million in China, with 80% living in rural villages [55].

Grandparents play a vital role in caring for LBC in rural areas, which can pose significant challenges. The National Sampling Survey in rural China showed that about 90% of grandparent caregivers of LBC aged over 50 and no more than 6% of grandmothers were educated more than primary school [56]. Based on a study of the elderly in rural China, about 6% of rural grandparent caregivers reported that they could barely provide child care due to poor health and 27% reported the child care to be challenging [56]. As a result, the relatively poor care given by some grandparents and the deprivation of parental care could cause delays in the health and development of rural LBC. In one study of this issue, Yue et al. investigated the impact of parental migration on young children from 6 to 30 months in 11 poor counties in rural China and found that parental migration during the earliest periods significantly reduced child cognitive development [57]. This delay in child development was largely mirrored by the reduction in dietary diversity and engagement in stimulating activities. Zhang et al. came to a similar conclusion using the nationally representative data from Chinese Family Panel Studies: grandparent care is strongly associated with delays in early child development, which is particularly disadvantageous for LBCs living in rural areas [58].

### 2.4. Lack of Access to Quality Healthcare

Access to quality health care could be another determinant of health status, as rural children with health conditions may receive less effective treatment due to poor access to quality healthcare. Although access to health services and utilization has improved significantly in recent decades, major disparities persist between children in urban and rural areas. While recent studies suggest that utilization of outpatient services are comparable, there are large gaps in the use of inpatient services between rural and urban residents as well as wealth gradients in both rural and urban areas [59,60,61,62].

Few studies directly compare the quality of health services in rural and urban areas, but available evidence suggests significant disparities in the quality of outpatient care. Several studies utilizing unannounced visits by standardized patients and clinical vignettes show large deficits in care in rural clinics and hospitals [63,64,65,66,67]. Based on one study conducted in three provinces, Shi et al. found that village clinicians completed only around 20% of diagnostic checklist items recommended by national guidelines when consulting a case of diarrhea in a young child [68]. The same study also found that only 9% of village clinicians and 14% of doctors in township hospitals correctly treated these patients, and providers were more likely to prescribe unnecessary or potentially harmful drugs. Though not directly comparable, a similar study conducted in an urban area evaluating care for standardized patients presenting with angina and asthma found that doctors in urban community health centers completed 32% of diagnostic checklists on average and correctly treated cases 24% of the time [69].

### 2.5. Poor Caregiving Health-Related Behavior

The poor health of rural children may also be due to the poor caregiver health-related behaviors among rural caregivers. Family choices regarding how to feed young children, how often a child sees a doctor, and whether to receive the immunization could have short-term and long-term health implications for children. Several studies in rural China have identified caregiving behaviors in child feeding as a factor in undernutrition and anemia among rural infants and toddlers [46,47,70,71,72,73]. For example, although public health practitioners recommend exclusive breastfeeding for the first six months after birth and continued breastfeeding for the first two years [74], studies have found low rates of exclusive breastfeeding under six months of age as well as over-reliance on formula among rural families. Reduced breastfeeding has also been linked to higher incidence of infant illness in rural areas of China [71].

In addition to inadequate breastfeeding practices, studies have also identified the late introduction of complementary foods, inadequate dietary diversity in complementary feeding, and limited use of micronutrient supplementation among caregivers of young rural children [46,71,72,73]. Inadequate dietary diversity is of particular concern. According to several studies, large shares of caregivers frequently do not provide children with the minimum level of dietary diversity, and micronutrient-rich foods (vegetables, fruits or meat) are provided less frequently than starchy staples such as grains [70,73,75]. These studies have also linked inadequate dietary diversity to micronutrient deficiencies, lower growth and developmental delays among infants and toddlers in rural China [46,70,75].

There are several socioeconomic factors that may contribute to inadequate child feeding practices. For example, maternal out-migration (leaving infants in the care of grandparents) may be one contributing factor in the low rates of breastfeeding in rural China. The extensive commercial promotion of formula in both rural and urban areas may also contribute to diminished breastfeeding rates and over-reliance on formula. Studies have also pointed to lack of information as the primary factor in inadequate feeding behaviors. Studies using both quantitative and qualitative methodologies have found that large shares of rural caregivers are unfamiliar with the causes of nutritional deficiencies such as anemia, their relation to child health and development, and how to adequately provide their children with healthy and complete diets [72,73]. Yue et al. also noted that few rural caregivers have reliable sources of information on child feeding, and many rely instead on their own intuition or previous childrearing experiences [73].

There is also some evidence of issues in health care utilization by rural caregivers. Studies have found improper use of antibiotics to be relatively common, with many rural caregivers deviating from doctor instructions in giving their children antibiotics or self-medicating children with antibiotics in lieu of seeing a doctor [76,77]. Studies examining health care utilization have identified poverty, education, and ethnicity as primary factors in caregiver behavior. Several research teams in China’s rural areas have found poverty and lower education levels to be associated with reduced use of maternal and child health services [42,43,78,79,80], and inaccurate knowledge regarding antibiotics has been associated with improper antibiotic usage among caregivers [76,77]. Ethnic minority households, especially those who lived in remote rural regions, were also found to use fewer maternal and child health services and have lower rates of child immunization compared with Han women, possibly due to having lower levels of education, higher prevalence of poverty, and higher tendency to follow local norms rather than following nationally/provincially recommended services and practices [42,43,45,78,79,80,81,82].

A handful of studies have examined the impacts of interventions designed to improve socioeconomic factors, as well as caregiver knowledge and attitudes, on child health and development [83,84,85,86,87,88,89]. Conditional cash transfer programs seem to only have modest effects on improving the knowledge of rural caregivers and the adoption of optimal practices without combining with a health education component [83,84]. Health knowledge interventions, however, have generally been found effective in increasing child immunization coverage and improving child health and development [85,86,87,88,89]. Few studies have examined effective interventions for improving infant and child feeding practices: only two small-scale interventions in urban cities have explored the positive effects of education interventions on promoting exclusive breastfeeding [90,91]. Evidence is still lacking in rural areas; however, the documented associations between low nutrition knowledge and inadequate feeding behaviors suggest that knowledge interventions may be an effective solution [72,73].

### 2.6. Environmental Determinants

Finally, geographic living environments and community resources may also contribute to the health gaps of rural children. Access to clean water and sanitation is one of these factors. China has implemented a rural drinking water program since the 1980s to increase access to safe drinking water for rural families. However, only 55% of rural households had access to on-premises tap water by 2015 [92]. Living in a place with access to tap water is correlated with the family’s socioeconomic status and possibly has long-term implications for child development. Chen et al. examined the impact of early life exposure to tap water and found a significant effect on children’s cognitive development in a sample of rural children aged 10–15 [93].

Another leading environmental factor that affects rural children is injury. Xiong et al., using surveillance data during 2009–2014, reported that injury has become one of the leading causes of death among children younger than five, with injury mortality in rural areas 3.73 times higher than the urban areas. Drowning accounted for 43.63% of these injury-related deaths [94].

### 2.7. Summary

Figure 1 presents a graphical representation of the major factors examined in this review that contribute to the health gaps of rural children. We note, however, that child health outcomes are affected by a wide range of complex behavioral, social, and biological factors. Although many of these factors have been identified in the existing literature, many remain unexplored or are unobservable.

## 3. Conclusions

In this paper, we discussed major factors that have likely contributed to the poorer levels of health among rural children in China. In addition to the direct income effect on children’s health, we summarize available evidence documenting potential underlying causes of health disparities. Despite significant progress in recent decades, children in rural areas face disadvantages compared with their urban counterparts from the beginning of life: Prenatal care and infant health outcomes are worse in rural areas; caregivers have poor health outcomes and lack knowledge and support to provide adequate nurturing care to young children; there are large disparities in access to quality health care between rural and urban areas; and rural families are more likely to lack access to clean water and sanitation.

Certainly, strengthening the pediatric training and workforce in rural areas is essential to addressing the health gaps of rural children. Insufficient knowledge and diagnostic capacity of rural health providers pose significant challenges to the delivery of quality health care for rural children. However, this review indicates that inadequate access to quality health care is just one factor contributing to the inequities in urban/rural child outcomes. Other potential interventions include addressing the health needs of mothers and grandparent caregivers, improving parenting knowledge and nurturing care, improving access to clean water and sanitation for remote families, and, most importantly, targeting poverty itself.

Overall, this review highlights myriad sources of the observed health–wealth gradient. While we use China as a case study given its high levels of socioeconomic inequality, similar factors contribute to this gradient elsewhere. To inform policies that improve health outcomes for the poor, there is a critical need for research that moves beyond well-documented correlations and risk factors and identifies the causal drivers of health outcomes among children, particularly those in economically disadvantaged environments. Given the evidence that health outcomes in childhood have wide-ranging effects throughout the life course, policies effectively improving the health of children of low socioeconomic status are likely to have substantial returns.

## Figures and Tables

**Figure 1 ijerph-19-13845-f001:**
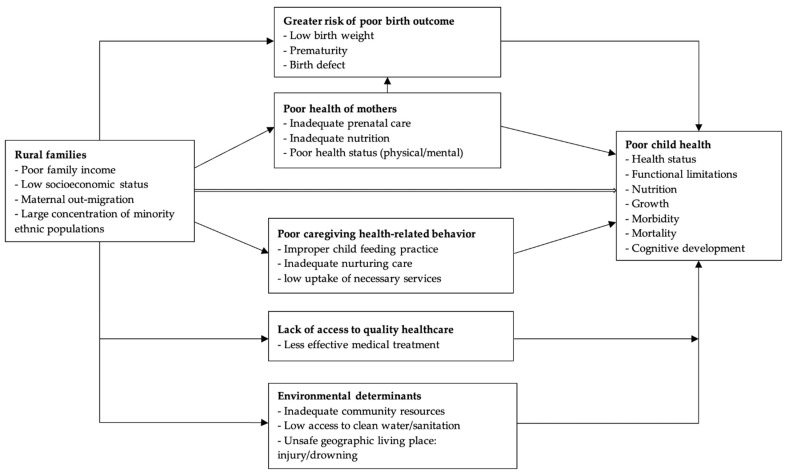
Review of major factors explaining the health gaps of rural children in China.

## Data Availability

Not applicable.

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
