# Peer review of "Structural Determinants of Child Health in Rural China: The Challenge of Creating Health Equity"

_ijerph, 2022, doi:10.3390/ijerph192113845_

Round 1

Reviewer 1 Report

An extremely well written literature review that contributes to a better understanding of the intersection between place, income and health equity among children in China, while engaging the broad international literature exploring these dynamics in other geographic settings. Below are my specific recommendations for further enriching the article.

Pg. 2, lines 85 and 86: It would be helpful to contextualize the decline in rural infant mortality rates (which is profound) to global or international rates for adding context to how significant this decline experienced in China has been.

Page 3, Section 2.1. Rebecca Blank's framework of cumulative disadvantage/cumulative discrimination maybe a good theoretical frame to introduce the complexity of causality and dynamic nature of the factors influencing health outcomes/income and place. Citation below.

Blank, Rebecca, M. 2005. "Tracing the Economic Impact of Cumulative Discrimination." American Economic Review, 95 (2): 99-103. DOI: 10.1257/000282805774670545

Page 4, Section 2.2. Integrating the framework of life course theory applied to maternal/infant health maybe a helpful theoretical construct to add to this section addressing infant health (see literature reference by Lu and Halfon below).

Lu, M. C., Kotelchuck, M., Hogan, V., Jones, L., Wright, K., & Halfon, N. (2010). Closing the Black-White gap in birth outcomes: a life-course approach. Ethnicity & disease, 20(1 Suppl 2), S2–76.   Halfon, N., Larson, K., Lu, M., Tullis, E., & Russ, S. (2014). Lifecourse health development: past, present and future. Maternal and child health journal, 18(2), 344-365.   Page 6, line 285: This was the first reference to ethnic differences within the rural population, is this the only study documenting ethnic differences or is there additional literature (and other ethnic groups) that could be integrated to discuss this topic?   Final comment: A graphic representation of the inputs and other variables impacting rural child health specifically would be an important addition to your conclusion. I would recommend a graphic or figure that allows readers to quickly see the large volume of variables you have explored in the literature review.

Reviewer 2 Report

Dear authors, congratulations to your work.

1. I would suggest you to think about purpose of your study. "The purpose of this paper is to identify some of the major factors that have led to the relatively poorer levels of health among China’s rural children". Your paper represents the review of papers already published and their results. According to that, you should aimed to "make a review of and list the major factors that have led to the relatively poorer levels of health among China’s rural children".

2. I would recommend you to change the subtitle "Discussion" to a "Conclusion"  

Reviewer 3 Report

Thank you for the opportunity to review the manuscript “Structural Determinants of Child Health in Rural China: Challenge of Creating Health Equity” for International Journal of Environmental Research and Public Health. Overall, it was thought provoking and enjoyable read. Generally speaking, I only have positive things to say about this research and don’t have any substantive concerns regarding the manuscript. There is much to like with this review paper. The study is well positioned, the literature focused and appropriately reviewed, the discussion clear and points to new directions for research. I applaud the author for undertaking such important work. My only recommendation would be for the authors to not refer to previous literature as having identified “causal” relationships. Not only is the notion of causality in social research patently observed, but the authors’ use of the term also downplays the importance of considering other factors that may influence child health. Making this adjustment in language would, in my view, would strengthen an already strong manuscript and broaden the contributions of this work to a larger literature.
